# Effects of Acute Fish Oil Supplementation on Muscle Function and Soreness After Eccentric Contraction-Induced Muscle Damage

**DOI:** 10.3390/nu17213408

**Published:** 2025-10-29

**Authors:** Sang-Rok Lee, Dean Directo, Yangmi Kang, Joshua Stein, Mason Calvert, Yong Woo An, Do-Houn Kim

**Affiliations:** 1Department of Kinesiology, Samford University, Birmingham, AL 35229, USA; 2Department of Kinesiology, New Mexico State University, Las Cruces, NM 88003, USA; ddirecto@nmsu.edu (D.D.); jwmstein@nmsu.edu (J.S.); mcalv505@gmail.com (M.C.); 3Department of Exercise Science and Athletic Training, Ithaca College, Ithaca, NY 14850, USA; ykang@ithaca.edu; 4Department of Health and Human Sciences, Loyola Marymount University, Los Angeles, CA 90045, USA; yongwoo.an@lmu.edu; 5Department of Human Physiology, Gonzaga University, Spokane, WA 99258, USA; kimd@gonzaga.edu

**Keywords:** fish oil, muscle damage, muscle function, muscle soreness, inflammation

## Abstract

Purpose: The primary aim of this study was to determine the efficacy of acute fish oil (FO) supplementation on indices of exercise-induced muscle damage (EIMD) in young healthy adults. Methods: Twenty-two healthy young male and females were randomly assigned to two experimental groups: fish oil (FO) or placebo control (CON). Participants performed a muscle damage protocol consisting of 10 sets of 10 plyometric drop jumps. Vertical jump height, isometric maximal voluntary contraction (MVC) torque, and systemic inflammation markers were assessed at pre-exercise, immediately post (post-0), post-24, post-48, and post-72 h. Results: Vertical jump performance and quadriceps peak torque significantly decreased in the CON group at post-0, 24, and 48 h (*p* < 0.05), while FO group recovered to baseline levels by post 48 h. Hamstring peak torque reductions recovered in the FO group at post-48 h but remained suppressed in the CON group until post-72 h (*p* < 0.05). Muscle soreness was significantly higher in the CON group compared to the FO group at post-48 h (*p* < 0.05). Systemic TNF-α levels significantly increased from baseline to post-0, 24, and 48 h in both groups (*p* < 0.05), with the CON group showing a trend toward incomplete recovery (*p* = 0.065). Conclusions: Our findings indicate that acute FO administration may modestly aid muscle recovery and reduce muscle soreness following EIMD in healthy young adults while the overall impact may be limited.

## 1. Introduction

It is well acknowledged that engaging in unaccustomed rigorous exercise, such as eccentric contractions (EC), may provoke exercise-induced muscle damage (EIMD) in individuals ranging from novice exercisers to highly trained elite athletes. EIMD can cause a loss of contractile force of skeletal muscle, reduced range of motion (ROM), and muscle swelling accompanied with delayed onset of muscle soreness (DOMS) [1,2]. DOMS typically peaks 24 to 72 h after EC [1], impairing muscle strength and flexibility, which can negatively impact training, exercise performance, and the ability to continue physical activity. These unfavorable outcomes may stem from micro-damage within the involved musculature [3], along with increased localized inflammation [4] and oxidative stress [5]. While the microtrauma of muscle fibers, localized inflammatory response, and production of reactive oxygen species (ROS) play adaptive roles in muscle remodeling and adaptation, mitigating the detrimental impacts of exercise-induced muscle damage (EIMD) is essential for optimizing recovery and performance, particularly for athletes facing multiple competitions in a short timeframe [6]. In addition, alleviating DOMS during the acute recovery period is crucial for promoting adherence to regular exercise among recreational exercisers, serving as a key psychophysiological factor in maintaining consistent physical activity [7].

Several strategies have been explored to expedite recovery and alleviate EIMD, with nutritional intervention, particularly those involving non-steroidal anti-inflammatory agents, emerging as effective approaches [8,9]. Among these, omega-3 polyunsaturated fatty acids (ω-3 FAs) have garnered significant attention for their health-enhancing properties. Comprised of eicosapentaenoic acid (EPA) and docosahexaenoic acid (DHA) derived from fish oil (FO), ω-3 FAs possess both anti-inflammatory and antioxidant properties [10,11], attenuating activities of inflammation and oxidative stress mediators. Moreover, ω-3 FAs have shown to enhance exercise performance, promote recovery from physical exertion, improve aerobic capacity, and support immune function [12,13,14].

Preliminary evidence suggests that FO supplementation enriched with ω-3 may effectively mitigate indices of EIMD such as DOMS and inflammation [15]. However, findings remain inconsistent, some studies have reported improved recovery of muscle function and reduced soreness following FO supplementation, while others have shown minimal or no benefits. These discrepancies may be partly attributed to variations in intervention programs such as varying differences in ω-3 dosage, duration of administration, exercise protocol for muscle damage, and dependent variables (e.g., physical function) [16,17,18].

The primary aim of the present study was to evaluate the efficacy of acute FO supplementation on skeletal muscle function (vertical jump performance, isometric peak torque of the quadriceps and hamstrings) following EIMD. The secondary aim was to assess changes in muscle soreness and systemic inflammation [tumor necrosis factor-alpha (TNF-α) and interleukin-6 (IL-6)] during the recovery period. This study aimed to address gaps in the current knowledge by determining whether short-term FO supplementation could enhance recovery and reduce inflammation in physically active young adults following intense lower-limb eccentric exercise. We hypothesized that FO administration would enhance recovery by improving indices of EIMD, and reducing inflammation, thereby promoting faster restoration of muscle function.

## 2. Materials and Methods

### 2.1. Participants and Experimental Design

Twenty-two healthy young males and females (*n* = 22; 22.13 ± 3.87 y) participated in this study. Inclusion criteria required participant to: (1) be healthy without any physical or mental disorders; (2) be non-smoker; (3) have no prior use of ω-3 FAs supplements; (4) not use anti-inflammatory drugs; (5) not engage in resistance exercise training; and (6) consume no more than five alcoholic drinks per week. All prospective participants underwent a screening process to confirm eligibility. Those meeting all criteria provided written informed consent prior to participating in the study. The study protocol was approved by the New Mexico State University Institutional Review Board for Human Subject (approval no. 19647).

The study implemented a randomized, double-blinded, placebo-controlled design that was stratified by sex and body weight to ensure comparable characteristics between groups. Participants were instructed to maintain their regular dietary habits and refrain from resistance exercise outside of the study protocol. Each participant visited the laboratory a total of 4 times. The participants were allocated to either a fish oil supplementation (FO, *n* = 11) or a placebo control group (CON, *n* = 11). During their first visit, baseline blood samples were collected, followed by the EIMD protocol, which was conducted only once. Immediately after the protocol, post-exercise blood samples were collected. Subsequent laboratory visits occurred at 24, 48, and 72 h post-EIMD, during which blood samples, physical function tests, and perceived muscle soreness assessments were conducted (Figure 1).

### 2.2. Muscle Damaging Exercise Protocol

To induce EIMD, participants completed 10 sets of 10 plyometric drop jumps with 10 s of rest between each jump and 1 min of rest between each set. For each jump, participants stepped off a 0.4-m-high box, landed on both feet, immediately performed a maximal vertical jump, and landed in a squat position with approximately 90 degrees of hip and knee flexion. This procedure was consistently followed for all 10 sets.

### 2.3. Fish Oil Supplementation

The FO supplement comprised a proprietary blend containing 700 mg of eicosapentaenoic acid (EPA) and 240 mg of docosahexaenoic acid (DHA) per capsule. The FO group consumed 3 FO capsules daily, one with each meal, for a total daily intake of 2100 mg EPA and 720 mg DHA during the experimental period. The place CON group received identical-looking soybean placebo capsules at the same dosage. Participants were instructed to consume the capsules as directed and return any unused or empty capsules to monitor their compliance. All participants were instructed to maintain their habitual diet and to avoid consuming food rich in omega-3 fatty acids.

### 2.4. Physical Function Assessment

#### 2.4.1. Vertical Jump Test

A vertical jump test was performed by a single technician using a vertical jump tester (Vertec, Jump USA, Sunnyvale, CA, USA). The standing reach was first measured by having participants stand with their both feet flat on the ground, keeping their legs and torso straight, and reaching upward as far as possible. The tester recorded the standing reach. Participants then performed a vertical jump, tapping the Vertec markers at the highest point of their jump. The vertical jump height was calculated by subtracting the standing reach from the highest jump reach. Each participant was allowed three attempts, with the highest jump recorded.

#### 2.4.2. Maximal Voluntary Contraction Measurements

Maximal voluntary isometric contraction (MVIC) strength of quadriceps and hamstring muscles were assessed using an isokinetic dynamometer (BiodexTM, Shirley, NY, USA). Participants were positioned on the dynamometer with their knee at 90 degrees and performed three 5-s MVC trials for the quadriceps (knee extension) and hamstring (knee flexion), with 30 s of rest between each trial. The highest peak torque recorded was used for analysis.

#### 2.4.3. Perceived Muscle Soreness Measurements

Perceived muscle soreness was assessed using a 10-cm visual analogue scale (VAS), ranging from 0 to 10. A score 0 indicates “no pain”, while a score 10 represents “excruciating pain”. Participants were instructed to perform an unweighted squat to approximately a 90-degree knee angle for 3 s and then mark their perceived soreness level on the VAS at each time point.

#### 2.4.4. Blood Collection and Analysis of Systemic Biomarkers

Blood samples were collected from each participant’s forearm vein using a 23-gauge needle and vacutainer tubes containing EDTA at each time point. The collected samples were promptly centrifuged to separate plasma and stored in multiple aliquots at −80 °C for analysis of inflammatory biomarkers. Plasma tumor necrosis factor-alpha (TNF-α) and interleukin-6 (IL-6) concentrations were assessed using commercially available ELISA kits (RayBiotech, Atlanta, GA, USA). All samples were analyzed in duplicate.

#### 2.4.5. Statistical Analysis

Statistical analysis was conducted using a SPSS software (version 31, IBM, Armonk, NY, USA). All data are presented as the mean ± standard deviations (SD). After confirming normality, homogeneity of variances, and sphericity, a mixed linear-model ANOVA was employed, with baseline MVC strength as a covariate to determine whether changes in dependent variables over the experimental period were attributable to sex, group, or a sex-by-group interaction. A Bonferroni correction was performed for multiple comparisons. When significant group x time interactions were noted, post hot analyses were performed to detect local differences. A criterion α-level for statistical significance was set at *p* ≤ 0.05. Sample sizes were determined by a priori power analysis using an effect size calculated from a previous study in which FO supplementation enhanced recovery of skeletal muscle function and muscle soreness [19]. Parameters for the power analysis included α = 0.05 and 1-β = 0.80.

## 3. Results

Descriptive data for research participants are provided in Table 1. There were no notable differences in age and anthropometric characteristics between groups.

### 3.1. Muscle Performance Recovery

The muscle function data are presented in Figure 1. There was a significant group x time interaction noted for vertical jump (F = 11.119, *p* < 0.001). Baseline vertical jump height values were similar for both FO and CON. Vertical jump performance was significantly decreased in the CON group at post-0 h (*p* < 0.001), -24 h (*p* < 0.001), and -48 h (*p* = 0.002). Vertical jump performance in FO was reduced at post-0 h (*p* < 0.001), -24 h (*p* = 0.003), and returned to the baseline levels at post-48 h. Vertical jump performance was significantly higher in FO compared to CON at post-48 h (*p* < 0.001)

A significant group x time interaction was found for quadriceps peak torque (F = 11.119, *p* < 0.001). In the CON group, quadriceps peak torque was significantly reduced at post-0 h (*p* = 0.001), -24 h (*p* < 0.001), and -48 h (*p* = 0.007). The FO group also presented substantially reduced quadriceps peak torque at post-0 h (*p* < 0.001), -24 h (*p* = 0.001), and returned to the baseline levels at post-48 h. The FO group presented a significantly greater quadriceps peak torque than the CON group at post-24 h and -48 h (*p* < 0.001).

A significant group x time interaction was noted for hamstring peak torque (F = 3.149, *p* = 0.019). Hamstring peak torque in the CON group was markedly decreased at post-0 h (*p* < 0.001), -24 h (*p* < 0.001), -48 h (*p* = 0.001), and -72 h (*p* = 0.003). The FO group also showed significantly reduced hamstring peak torque at post-0 h (*p* < 0.001), -24 h (*p* < 0.001), post-48 h (*p* < 0.001), but returned to the baseline levels at post-72 h. Hamstring peak torque was remarkably higher in FO than CON at post-24 h (*p* < 0.001).

### 3.2. Perceived Muscle Soreness

There was a significant group x time interaction for muscle soreness (F = 11.119, *p* < 0.001). The muscle soreness data are presented in Figure 1. Compared to baseline values, quadriceps muscle soreness in the CON group remarkably increased at post-0 h (*p* < 0.001), -24 h (*p* < 0.001), and -48 h (*p* < 0.001). In the FO group, muscle soreness was significantly increased at post-0 h (*p* < 0.001), -24 h (*p* = 0.003), and then returned to the baseline levels at post-48 h. At post-24 h, muscle soreness was significantly greater in the CON group than the FO group (*p* < 0.001). The within- and between-treatment outcomes for muscle function and muscle soreness recovery are presented in Table 2.

### 3.3. Biochemical Data

The blood inflammation data are presented in Figure 2. A significant main effect of time was noted for TNF-α (F = 36.539, *p* < 0.001), whereas no significant group or group x time interaction was found. TNF-α significantly increased in the CON group at post-0 h (*p* < 0.001), -24 h (*p* < 0.001), and -48 h (*p* = 0.001), with a trend toward incomplete recovery at post-72 h (*p* = 0.065). The FO group also exhibited a significant increase in TNF-α at post-0 h (*p* < 0.001), post-24 h (*p* < 0.001), and post-48 h (*p* = 0.021), but values returned to pre-EIMD levels by post-72 h. There was a significant time effect for IL-6 (F = 30.698, *p* < 0.001) while no group or group x time interaction was noted. IL-6 was markedly elevated in the CON group at post-0 h -24 h, and -48 h (*p* < 0.001). Similarly, the FO group presented a remarkable increase in IL-6 at post-0 h (*p* < 0.001), -24 h (*p* = 0.002), and -48 h (*p* = 0.008) (Figure 3).

## 4. Discussion

The present study investigated the efficacy of acute ω-3 administration through FO supplement on indices of EIMD and muscle function recovery following a single bout of vigorous eccentric contractions in healthy young adults. The key findings of this study demonstrated that acute FO supplementation ameliorated some indices of EIMD and muscle function recovery, supporting our hypothesis.

We observed the remarkable reduction in physical function; decreases in vertical jump performance, peak isometric torque of quadriceps and hamstring muscles and an increase in muscle soreness following rigorous eccentric contraction-induced exercise in young adults. It is widely recognized that unaccustomed, vigorous training, such as eccentric contraction, can lead to negative outcomes, including declines in muscular strength, range of motion, and significant muscle soreness [19,20]. These unfavorable effects may arise from the structural changes in damaged muscle fibers, along with the upregulation of inflammatory mediators, which can hinder muscle recovery, impair physical function, and ultimately reduce physical performance [21].

Our findings demonstrated a significant reduction in vertical jump performance in both the FO and CON groups immediately post-EIMD, which remained suppressed at post-24 h. However, acute FO supplementation appeared to mitigate this decrement. Specifically, the FO group exhibited vertical jump performance recovery within 5% of baseline by 48 h post-EIMD, whereas the CON group had not recovered fully even at 48 h. Notably, vertical jump performance in the FO group at post-24 h was comparable to the levels observed in the CON group at post-72 h. In addition, we observed a significant group difference in vertical jump performance at post-24. Thus, these findings highlight the potential that acute FO supplementation may mitigate the loss of vertical jump performance following EIMD. These findings align with those reported by Jakeman and colleagues who demonstrated that high-dose acute FO supplementation significantly improved vertical jump performance compared to a placebo group following EIMD [22]. This consistency highlights the potential role of FO supplementation in enhancing muscle recovery. The current data suggest that FO supplementation may ameliorate skeletal muscle impairment and promote recovery following EIMD.

FO supplementation also significantly mitigated the loss of isometric peak torque of quadriceps and hamstring muscles following EIMD. While both FO and CON groups experienced significant reduction in quadriceps peak torque, the FO group demonstrated recovery to within 5% of pre-EIMD values by 48 h, whereas the CON group still exhibited a significant strength loss of 8.9%. Similarly, hamstring peak torque was significantly reduced in both groups post-EIMD. However, the FO group recovered to baseline levels by 72 h, while the CON group had not fully recovered.

Evidence regarding the efficacy of ω-3 supplementation for muscle recovery after EIMD remains inconsistent. Kyriakidou and colleagues reported no improvement in maximal voluntary isometric contraction following 4 weeks of ω-3 FAs supplementation post-EIMD [16], which are in accordance with the findings by Gravina and colleagues who also observed no impact on leg muscle strength recovery after 4 weeks of ω-3 FAs consumption [23]. In contrast, Ochi and colleagues reported that 8 weeks of FO supplementation resulted in significantly higher maximal elbow isometric contraction torque at 24 h post-EIMD compared to a placebo group (90.5% vs. 76.8%) [20], Similarly, Tsuchiya and colleagues demonstrated that elbow MVC was significantly higher in the ω-3 FAs group than in the placebo group 2–5 days after EIMD [18]. These mixed results underscore the need for further research to clarify the efficacy of ω-3 FAs administration on muscle function recovery, particularly following high-intensity exercise.

It is well established that physical function declines significantly following EIMD, manifesting as reduction in peak power, jump performance, and muscle contraction velocity [24,25,26,27]. The loss of muscle function following EIMD would be attributable to the structural damage of myofibrils and Z-line streaming, compromised muscle membranes, disturbed neuromuscular junction, impaired calcium homeostasis, and excitation-contraction coupling failure in skeletal muscle [28]. Notably, ω-3 FAs, a key component of cell membranes, enhance cell membrane integrity [29,30], which is linked to cellular function [31]. A previous study reported that rats fed FO for 8 weeks showed higher skeletal muscle membrane concentrations and exhibited smaller reductions in muscle tension during repeated 10-min contraction bouts compared with rats fed saturated fat [30]. This suggests that ω-3 FAs administration enhances muscle cell membrane function, potentially mitigating reductions in muscle performance following intense exercise. Overall, these findings highlight the potential of acute FO supplementation to improve muscle performance recovery after EIMD.

We found a significant group-by-time interaction for perceived muscle soreness, highlighting the differential responses between the CON and FO groups following EIMD. Quadricep muscle soreness in the CON group significantly increased from baseline levels to post-0, 24, and 48 h. In contrast, the FO group experienced increased muscle soreness at post-0 and 24 h but returned to baseline levels by 48 h. Notably, quadriceps muscle soreness was significantly greater in the CON group compared to the FO group at post-24 h (*p* < 0.001), suggesting that FO supplementation may have mitigated the prolonged perception of muscle soreness commonly associated with EIMD. This attenuation in perceived soreness aligns with the recovery of muscle function metrics, such as vertical jump performance and peak isometric torque in the FO group.

The attenuation of muscle soreness observed in the FO group may be attributed to the membrane-stabilizing properties of ω-3 FAs, which are known to reduce the intensity and duration of muscle soreness [32]. This reduction in perceived soreness occurs by reducing the production of inflammatory cytokines and eicosanoids [33] that are responsible for activating peripheral nociceptors that facilitate pain and heighten nociceptor sensitivity [34,35]. Additionally, ω-3 FAs have been evidenced to modulate central sensitization and neuropathic pain by inhibiting the activation of the mitogen-activated protein kinase pathway [36,37] and by increasing neuroplasticity and neurogenesis [38,39] that act to reduce central sensitization. This is consistent with previous studies demonstrating that ω-3 FAs supplementation can attenuate DOMS and enhance recovery by modulating inflammatory pathways. For instance, Tartibian and colleagues reported that ω-3 FAs supplementation reduced post-exercise muscle soreness and inflammation markers, such as TNF-α, following intense exercise [40]. Consistent with these findings, we found that TNF-α levels significantly increase from baseline to post 0, 24, and 48 h in both groups. However, TNF-α in the FO group returned to within 5% of baseline by 72 h (*p* = 0.33), whereas the CON group exhibited a trend toward incomplete recovery, remaining 9.5% above baseline (*p* = 0.065). Moreover, ω-3 FAs supplementation downregulates the NF-κB (Nuclear Factor Kappa-light-chain-enhancer of Activated B Cells) pathway, an important pathway that is responsible for controlling genes involved in immune and inflammatory responses [41] as well as protein catabolism [42]. These findings support the potential anti-inflammatory effect of ω-3 FAs administration in alleviating DOMS and enhancing recovery following intense exercise. The limited statistical power for the inflammatory mediator in the present study may be attributed to the short duration and/or insufficient dosage of ω-3 FAs supplementation, which may not have allowed for adequate incorporation of ω-3 FAs into cell membranes. Future research should explore the optimal timing and dosage of ω-3 FAs supplementation to maximize its benefit for recovering inflammation markers and overall muscle recovery.

It is well established that DOMS primarily arises from mechanical disruption to muscle fibers and subsequent inflammation. The infiltration of immune cells, such as neutrophils and macrophages, into damaged muscle tissues contributes to the release of pro-inflammatory cytokines, leading to heightened pain sensitivity and soreness [43]. ω-3 FAs are known to modulate these inflammatory responses by incorporating into cell membranes and influencing the production of eicosanoids, key mediators of inflammation [44]. In the present study, the significant reduction in muscle soreness observed in the FO group could be attributed to its anti-inflammatory effects. However, we acknowledge that we did not assess underlying mechanisms directly, which is a limitation of this study. Future research should include direct mechanistic measures to confirm the anti-inflammatory actions of FO. By limiting the extent of secondary muscle damage, FO supplementation likely facilitated a faster resolution of muscle soreness. Additionally, the enhanced recovery of muscle function metrics in the FO group provides additional support for the hypothesis that FO supplementation promotes a more efficient recovery process.

While the findings of this study are promising, it is important to address the following limitations in the present study. First, dietary control was based solely on participant instruction rather than direct verification. Additionally, the study did not include direct assessments of tissue-level inflammation, which would have provided more robust evidence to elucidate the underlying mechanisms of the observed results. Future studies should aim to address these limitations by incorporating direct measures of inflammation and ensuring stricter control over participant compliance.

## 5. Conclusions

The novel contribution of the study lies in demonstrating that acute FO supplementation can attenuate specific detrimental effects of EIMD and enhanced early recovery of muscle function in healthy young adults. These findings provide compelling evidence supporting the efficacy of acute FO administration in promoting more efficient recovery and preserving physical function following strenuous exercise. Moreover, they highlight the potential of ω-3 FAs as a strategy to support muscle recovery, providing a foundation for future research to optimize intervention parameters and further improve physiological and functional outcomes in individuals undergoing intense exercise training.

## Figures and Tables

**Figure 1 nutrients-17-03408-f001:**
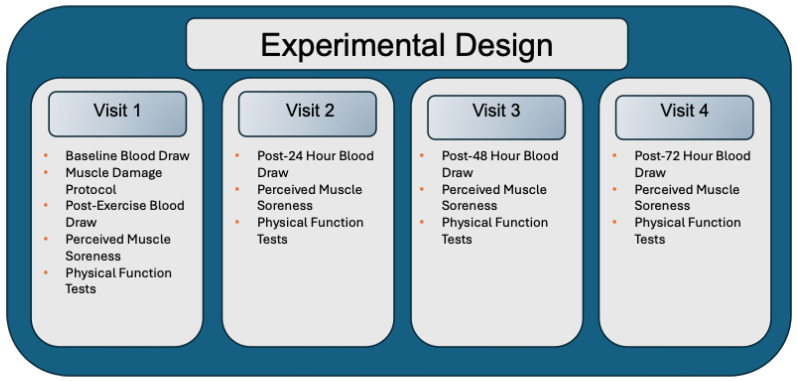
Experimental Design. During Visit 1, participants first underwent a baseline blood draw followed by the muscle damage protocol. Immediately after the protocol, a post-exercise blood sample was collected. Participants then completed assessments of perceived muscle soreness and physical function. During visits 2, 3, and 4 (24, 48, and 72 h post-EIMD, respectively), each session included a blood draw followed by assessments of perceived muscle soreness and physical function.

**Figure 2 nutrients-17-03408-f002:**
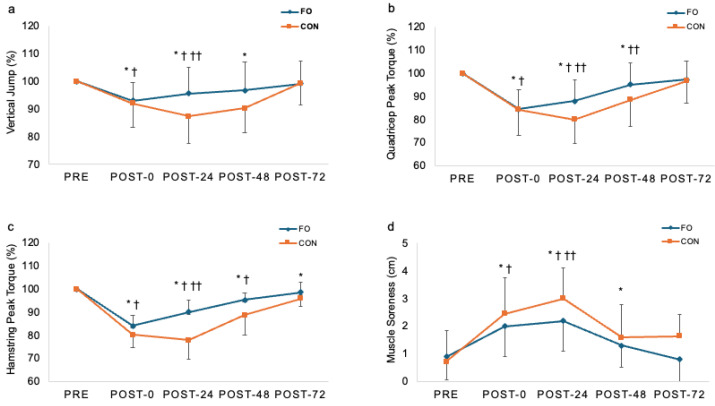
Changes (mean ± SD) in vertical jump performance (**a**), peak torque of quadriceps (**b**) and hamstring (**c**), and perceived muscle soreness (**d**) at post-0, post-24, post-48, and post-72 h after eccentric contraction. FO: Fish oil; CON: placebo control. * *p* ≤ 0.05, difference from pre-exercise levels in CON; ^†^ p ≤ 0.05 difference from pre-exercise levels in FO, ^††^ p ≤ 0.05, difference between groups.

**Figure 3 nutrients-17-03408-f003:**
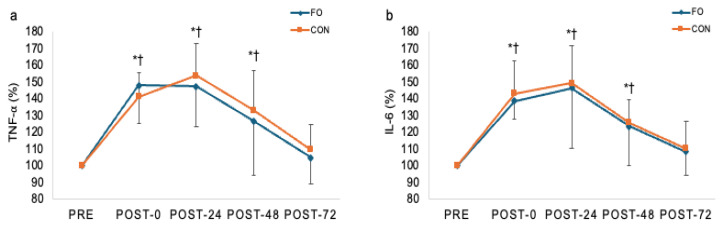
Changes (mean ± SD) in TNF-α (**a**) and IL-6 (**b**) at post-0, post-24, post-48, and post-72 h after eccentric contraction. FO: Fish oil; CON: placebo control. TNF-α = Tumor Necrosis Factor-Alpha; IL-6 = Interleukin-6. * *p* ≤ 0.05, difference from pre-exercise levels in CON; ^†^ *p* ≤ 0.05 difference from pre-exercise levels in FO.

**Table 1 nutrients-17-03408-t001:** Descriptive characteristics of subjects at baseline. FO = fish oil; CON = placebo control.

	FO	CON
Age (years)	23.3 ± 4.5	20.6 ± 1.1
Height (cm)	170.3 ± 6.3	171.0 ± 10.4
Weight (kg)	71.2 ± 11.8	69.9 ± 12.1
Body mass index (kg/m^2^)	24.6 ± 4.5	23.8 ± 2.1

**Table 2 nutrients-17-03408-t002:** Changes in muscle function and muscle soreness following exercise-induced muscle damage.

Outcome Measure	Group	Baseline	Post-0 h	Post-24 h	Post-48 h	Post-72 h	Time Effect	Group Effect	Time × Group Interaction
Vertical Jump (cm)	FO	49.42 ± 8.7	43.1 ± 7.4	43.9 ± 8.8	46.7 ± 9.3	48.4 ± 10.1	*p* < 0.001	*p* = 0.933	*p* = 0.002
	CON	50.1 ± 13.1	42.4 ± 10.4	40.3 ± 13.2	44.3 ± 12.4	48.9 ± 14.4			
Quadriceps Peak Torque (Nm)	FO	177.7 ± 39.1	150.2 ± 37.3	157.2 ± 42.9	168.4 ± 37.3	173.0 ± 41.5	*p* < 0.001	*p* = 0.507	*p* = 0.001
	CON	172.3 ± 50.1	146.5 ± 51.4	139.9 ± 50.9	153.7 ± 52.8	167.8 ± 57.0			
Hamstring Peak Torque (Nm)	FO	72.8 ± 16.0	61.6 ± 15.6	65.8 ± 15.5	72.0 ± 15.6	72.0 ± 15.6	*p* < 0.001	*p* = 0.465	*p* = 0.019
	CON	71.8 ± 17.4	57.8 ± 15.7	56.3 ± 16.2	64.0 ± 18.0	66.4 ± 18.7			
Muscle Soreness (Nm)	FO	0.9 ± 0.9	3.6 ± 1.4	4.0 ± 1.3	1.3 ± 1.3	1.1 ± 1.6	*p* < 0.001	*p* = 0.038	*p* < 0.001
	CON	0.7 ± 0.7	3.8 ± 1.5	4.8 ± 1.2	3.7 ± 1.4	1.3 ± 1.1			

## Data Availability

The data presented in this study are available on request from the corresponding author. The data are not publicly available due to privacy.

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
