# Peer review of "Effects of Acute Fish Oil Supplementation on Muscle Function and Soreness After Eccentric Contraction-Induced Muscle Damage"

_nutrients, 2025, doi:10.3390/nu17213408_

Round 1
Reviewer 1 Report
Comments and Suggestions for Authors
The authors present a well-designed study and well-written manuscript according to the design and current data analysis.
However, the effort in design, conduct, and writing up is compromised by the statistical model chosen.
At this point in the review, I recommend that the data are reanalysed using a more appropriate statistical model. Secondly, a sample size justification should be provided. Thirdly, an indication of smallest important effect size should be included within the manuscript and inferences based around that; Cohen d is statistically useful as provides a measure of effect size relative to sample variation, but it's not always best for performance recovery studies where metrics are available for smallest important effects for sports. e.g. Hopkins et al. 1999 MSSE. Design and Analysis of....
Specific suggestions for data reanalysis:
- The repeated-measures ANOVA assumes independence at each sample point. This is wrong as there are correlated data.
- Testing each time point induces incredible type-2 error rates; you are missing effects.
- Testing multiple time points induces inflated type-1 error rates.
- Cohen d, you must use the baseline SD not the pooled SD. It's change from baselinet that matters as the reference effect size. At least didn't use SD of the change score, which is a disaster.
Suggested approach:
- Mixed linear model. While you could apply a repeated-measures analysis, you'd have to address the correlations between time points with a unstructured random effects matrix. e.g. in SAS repeated time/type=un; random participant;
- You need Period in the fixed effect model to accound for order effect.
- Much of the time-series is presented as curvy-linear and the research question appears to be overall effect on recovery with time-series as secondary (while the current write-up places time-series as primary and overall treatment effect as secondary, which likley to be the bias from significance testing). Therefore, I suggest changing to a numerial linear model approach. A 1st order polynomial would be appropriate to address the curve in the time-series data, with time as a numeric effect to get an overall effect.
- Grand-mean centre data.
- Yes, apply baseline value as a covariate.
- You may get better model fit for the parametric data with log-transformation.
- Random effects: participant.
Next, data presentation.
Presenting a lot of outcome statistics in the text makes the Results difficult to read.
If you move to the linear model approach suggested above, then you will only overall data. This can all be compiled for your within treatment (time effect from numeric linear component) and between treatment outcomes in a Results Table to accompany the figures. The Results text can then be simplifed with reference to the stats outcomes in the Table. You would also need to change the statistics symbols in the figures.
To obtain time point effects from the linear model with time as a numeric, you can estimate the effect at a point of time you are interested in from the model by specification within the statistics model. See a statistician if unsure how to do this. You don't necessary need to present estimates for each sampling point; you can select say mid time point and end time point only. This reduces type-1 error amplification.
Improving the statistical analysis should improve this paper and may draw out the nutritional effect.
Author Response
Dear Reviewer,
We sincerely appreciate your thorough evaluation of our manuscript and the insightful comments you provided. Your feedback has greatly strengthened the clarity, rigor, and overall quality of our work. Please find attached our detailed responses to all comments.

Reviewer 2 Report
Comments and Suggestions for Authors
This manuscript is Important because it addresses the area of acute fish oil supplementation and its effects on exercise-induced muscle damage (EIMD). While chronic omega-3 intake has been widely studied, acute supplementation remains less understood, and the current study provides valuable evidence suggesting potential recovery benefits. The study is a randomized, double-blind design, it has objective measures of muscle function, and the inclusion of inflammatory markers.
Detailed Comments by Section
Abstract
Line 14–29: The abstract overstates the positive effects. The interpretation should be more cautious.
The phrase “attenuating the elevation of inflammation” is misleading since both groups showed significant increases in TNF-α and IL-6 (lines 181–190); attenuation was not clearly demonstrated.
Specify that improvements were modest and not consistently significant across all markers.
Introduction
Lines 59–63: Prior findings are described as “conflicting” without citing specific differences in design (e.g., doses, exercise types).
Materials and Methods
Lines 99–101: I think there is a typo because it says 'and 720 g (!)' instead of 'DHA'. (it should be mg, not g
Randomization is described but method (e.g., block randomization, stratification by sex) should be specified
Only 22 participants were included. The author should include the sample size and power calculation.
The statistical methods do not describe correction for multiple comparisons despite repeated ANOVAs across several endpoints. This increases the risk of type I error.(use Bonferroni, Tukey or another method for multiple comparison correction)
Results
Line 158–159: There is a contradiction between the results and the discussion. In the result Vertical jump, it's said “no main effect of group or group × time interaction” is reported, while in the discussion (lines 207–216), it is interpreted as a meaningful difference. This inconsistency should be clarified.
Discussion
The discussion infers group differences in vertical jump recovery despite non-significant interactions. This overinterpretation should be toned down
The mechanistic explanation for reduced soreness is plausible, but direct evidence from this study (e.g., incorporation of ω-3 into muscle membranes) is missing. The authors should acknowledge this limitation.
Limitations are addressed; however, dietary intake (omega-3 from habitual diet) was not assessed, which could confound the results.
The discussion frequently cites studies with chronic supplementation (e.g., 4–8 weeks), which may not be directly comparable to acute administration.
Author Response

(The authors gave the same response as above.)

Reviewer 3 Report
Comments and Suggestions for Authors
Dear Authors,
The manuscript is engaging and addresses a topic of practical relevance. Please find my comments and suggestions below.
The abstract is overly lengthy; I recommend condensing it to focus more sharply on the crucial findings.
The introduction is well-written but could be enhanced in several areas. For instance, the reference to "conflicting results" in line 60 should be substantiated with greater precision and detail. Furthermore, the primary and secondary objectives should be delineated more accurately, with a clear description of the outcomes of interest. The concluding paragraph of the introduction should effectively establish the study's context and rationale, clearly articulating its necessity.
Within the Materials and Methods section, the experimental design is described rather superficially. Providing a more detailed account would significantly strengthen the manuscript. I also advise integrating a figure that provides a schematic overview of the experimental design. In general, providing a more detailed account by adhering to the CONSORT flowchart guidelines would significantly strengthen the manuscript.
A justification for the sample size is absent and should be included.
Line 98: The indicated dosage of DHA is 20 mg. Please verify if this is correct, as it may be a typographical error.
The discussion is clearly structured and comprehensive. However, several aspects could be refined to improve the overall impact. Certain proposed mechanisms are discussed only superficially.
In lines 276-277, the properties of omega-3 fatty acids warrant a more detailed explanation (please see: Therdyothin and Phiphopthatsanee, 2025. The Effect of Omega-3 on Mitigating Exercise-Induced Muscle Damage).
In lines 300-301, regarding the inflammatory effects, the NF-κB pathway at the muscular level should be mentioned (please consider Paoli et al., 2024. Not Only Protein: Dietary Supplements to Optimize the Skeletal Muscle Growth Response to Resistance Training: The Current State of Knowledge).
Finally, the novel contribution of this study should be highlighted more prominently to distinguish it from existing literature on the subject. Additionally, please review the bibliography, as the numbering appears to commence from reference number 7.
Author Response

(The authors gave the same response as above.)

Round 2
Reviewer 2 Report
Comments and Suggestions for Authors
Dear authors your manuscript has improved a lot. Please include in the discussion as limitation that Dietary control is based on participant instruction rather than verification (that is very comon but it is still a limitation)
Author Response
Comments 1: Dear authors your manuscript has improved a lot. Please include in the discussion as limitation that Dietary control is based on participant instruction rather than verification (that is very comon but it is still a limitation)
Response 1: We sincerely appreciate your suggestion. We have revised the discussion to include dietary control as a limitation, following the reviewer’s recommendation (lines 354–355).
Reviewer 3 Report
Comments and Suggestions for Authors
Well done. I suggest synthesizing conclusions. Also consider including Table 2 as supplementary material.
Author Response
Comment 1: Well done. I suggest synthesizing conclusions. Also consider including Table 2 as supplementary material.
Response 1: We sincerely appreciate your suggestion. We have synthesized the conclusions to combine the key points (lines 362–369). Table 2 has been provided in response to a request from another reviewer.